# Recent Advances in LDH/g-C_3_N_4_ Heterojunction Photocatalysts for Organic Pollutant Removal

**DOI:** 10.3390/nano13233066

**Published:** 2023-12-01

**Authors:** Cheng Du, Jialin Xu, Guixiang Ding, Dayong He, Hao Zhang, Weibao Qiu, Chunxue Li, Guangfu Liao

**Affiliations:** 1Shenzhen Institute of Advanced Technology, Chinese Academy of Sciences, Shenzhen 518055, China; duch3@mail3.sysu.edu.cn (C.D.); xujialin@mindray.com (J.X.); kaneh1994@163.com (D.H.); wb.qiu@siat.ac.cn (W.Q.); 2Shenzhen Mindray Bio-Medical Electronics Co., Ltd., Shenzhen 518000, China; zhanghao_e@mindray.com; 3College of Materials Engineering, Fujian Agriculture and Forestry University, Fuzhou 350002, China; dgx421148676@yeah.com; 4College of Ecological Environment and Urban Construction, Fujian University of Technology, Fuzhou 350118, China; chunxueli@fjut.edu.cn

**Keywords:** layered double hydroxides, carbon nitride, heterojunction, photocatalysis, organic pollutant removal

## Abstract

Environmental pollution has been decreased by using photocatalytic technology in conjunction with solar energy. An efficient method to obtain highly efficient photocatalysts is to build heterojunction photocatalysts by combining graphitic carbon nitride (g-C_3_N_4_) with layered double hydroxides (LDHs). In this review, recent developments in LDH/g-C_3_N_4_ heterojunctions and their applications for organic pollutant removal are systematically exhibited. The advantages of LDH/g-C_3_N_4_ heterojunction are first summarized to provide some overall understanding of them. Then, a variety of approaches to successfully assembling LDH and g-C_3_N_4_ are simply illustrated. Last but not least, certain unmet research needs for the LDH/g-C_3_N_4_ heterojunction are suggested. This review can provide some new insights for the development of high-performance LDH/g-C_3_N_4_ heterojunction photocatalysts. It is indisputable that the LDH/g-C_3_N_4_ heterojunctions can serve as high-performance photocatalysts to make new progress in organic pollutant removal.

## 1. Introduction

Environmental pollution has made it extremely difficult for humanity to flourish sustainably [1,2,3,4,5]. The use of photocatalytic technology to harness clean and reproducible solar energy is widely considered to be the most effective answer to the issues [6,7,8]. An effective method for reducing pollution is to use photocatalytic technology to degrade organic pollutants by using solar irradiation [9,10]. In principle, photocatalysis is an oxidation-reduction process. The fundamental components of the photocatalytic process, photocatalysts, are crucial in driving the reaction and harvesting light [11,12,13,14,15]. Consequently, research into highly effective photocatalysts is crucial for the advancement of photocatalytic technology.

An organic semiconductor having triazine or heptazine as its main constitutional unit and a layer structure resembling graphite is known as graphitic carbon nitride (g-C_3_N_4_) [7,8,16]. Since the use of g-C_3_N_4_ in photocatalytic hydrogen synthesis was initially discovered [17], it has garnered considerable interest. Because g-C_3_N_4_ possesses a smaller band gap (approximately 2.7 eV) than traditional photocatalysts like TiO_2_ [18], it can be stimulated by visible light. Furthermore, g-C_3_N_4_ is competitive among other photocatalytic materials due to its excellent chemical stability, high thermostability, affordable raw ingredients, and straightforward production procedure [19,20,21,22]. However, there are numerous issues with the industrial uses of g-C_3_N_4_. Only blue and purple lights, with a wavelength of 460 nm, may pass through g-C_3_N_4_, resulting in a low solar energy utilization rate [23]. The redox capacity of g-C_3_N_4_ is reduced because of the rapid recombination of photocarriers. The specific surface area (SSA) of g-C_3_N_4_ is relatively tiny because of its bulk characteristics. These flaws prevent g-C_3_N_4_ from being developed further for photocatalytic uses. To increase the photocatalytic performance of g-C_3_N_4_, various techniques have been used, including decorating co-catalysts, doping elements (such as F, O, Ni, Fe, etc.), building nanostructures, creating heterojunctions [24,25], etc. To boost the separation of charge carriers in g-C_3_N_4_, creating heterostructures with different semiconductors is the most important technique. Because of the varying Fermi levels between various photocatalysts, when they come into contact with one another, charge carriers can travel across the semiconductors, which ultimately create an internal electric field (IEF) between them. The electric field allows the photoinduced electrons and holes to flow in a certain direction, thereby separating them [26,27].

A highly efficient method for improving photocatalytic performance has recently been discovered by creating layered double hydroxides (LDHs)/g-C_3_N_4_ heterojunctions. Essentially, LDHs are a class of two-dimensional (2D) hydrotalcite-like clay materials made up of exchangeable interlayer anions and positively charged host layers [28,29]. LDHs and their derivatives have been found useful in a variety of sectors, especially photocatalysis, because of their inexpensive features, excellent chemical stability, customizable composition, homogeneous distribution of cations and anions, and interchangeable interlayer anions. Nevertheless, because of the fast recombination of photocarriers, the photocatalytic activity of single LDHs is unsatisfactory [30]. Therefore, by building LDH/g-C_3_N_4_ heterojunctions with a smart design, it is possible to obtain optimal photocatalysts with top performance while overcoming the drawbacks of g-C_3_N_4_ and LDHs.

Until now, although there have been numerous encouraging reviews on g-C_3_N_4_ and LDH-based photocatalysts [7,26,27,29,31,32,33,34], a comprehensive review specifically focusing on LDH/g-C_3_N_4_ heterojunctions for organic pollutant removal is unavailable. In this review, recent developments in LDH/g-C_3_N_4_ heterojunctions and their applications for organic pollutant removal are carefully reviewed. The advantages of LDH/g-C_3_N_4_ heterojunctions are first summarized to provide some overall understanding of them. Then, a variety of approaches to successfully assembling LDHs and g-C_3_N_4_ are simply illustrated. Last but not least, certain unmet research needs for the LDH/g-C_3_N_4_ heterojunction are suggested. This review can provide some new insights for the development of high-performance LDH/g-C_3_N_4_ heterojunction photocatalysts. It is indisputable that the LDH/g-C_3_N_4_ heterojunctions can serve as high-performance photocatalysts to make new progress toward organic pollutant removal.

## 2. Advantages of LDH/g-C_3_N_4_ Heterojunctions

It has been suggested that one efficient method to enhance the photocatalytic activity of LDH/g-C_3_N_4_ composites is to build 2D/2D heterojunctions. In addition to providing a large number of surface active sites to create heterojunctions, the 2D structure of LDHs and g-C_3_N_4_ significantly reduces the distance over which photocarriers must move within the 2D/2D heterojunctions, favoring a photocatalytic reaction [35,36]. LDHs are ideal materials for creating 2D/2D heterojunctions with g-C_3_N_4_ because of their suitable band structure and variable composition [29]. As is known to all, the band gap of LDHs is about 2.0–3.4 eV, which can be adjusted by changing and regulating the M^2+^ and M^3+^, which is advantageous for the capture of visible light [37]. In addition, the materials’ abundance of basic sites makes them useful as heterogeneous photocatalysts for various chemical processes [38]. The active sites of LDH/g-C_3_N_4_ 2D/2D heterojunctions are also changeable due to the ability to manipulate the cations and anions [39]. Moreover, designing LDHs’ interlayer space, number of layers, and functionalization with g-C_3_N_4_ are all rather simple processes (Figure 1). There are several advantages to LDH/g-C_3_N_4_ heterojunctions as shown below: (i) Owing to the close contact between LDHs and g-C_3_N_4_, photocarriers can conveniently migrate, which is beneficial for photocarrier separation [40]. (ii) The high surface area in LDH/g-C_3_N_4_ heterojunctions and their strong light harvesting ability are beneficial for photocatalytic activity [41]. (iii) LDH/g-C_3_N_4_ heterojunctions exhibit a shorter migration distance than photocarriers, thus reducing the electron-hole recombination [42]. (iv) The band structure can be simply adjusted [43]. All of these advantages ensure them with huge application potential in organic pollutant removal.

Based on the band alignment between the conduction band (CB) and the valence band (VB) of LDHs and g-C_3_N_4_, the heterojunctions formed between them can be classified into three types: straddling-gap junctions (Type I), staggered-gap junctions (Type II), and broken-gap junctions (Type III). When two semiconductors with staggered band alignment are intimately contacted, and there exists a charge migration between the CB in LDHs and the VB in g-C_3_N_4_, a Z-scheme system is formed. This Z-scheme system is characterized by an efficient electron-hole separation process, leading to enhanced photocatalytic activity. The charge transfer between the semiconductors results in the formation of a potential difference at the interface, driving the separation of photogenerated charge carriers and reducing their recombination rate. This mechanism significantly improves the overall efficiency and performance of the heterojunction photocatalysts. Therefore, understanding the band alignment and charge transfer processes in heterojunction systems is crucial for the design and optimization of advanced photocatalytic materials with superior performance. Although both the Z-scheme and Type II heterojunctions have the same staggered band alignment, e^−^-h^+^ transfer occurs in opposite directions in these two semiconductors. In contrast to Type II heterojunctions, where photogenerated e^−^ and h^+^ are transferred from high potentials to low potentials, leading to compromised redox capacities, a Z-scheme heterojunction can drive the photogenerated e^−^ in LDHs to migrate and then recombine with the photogenerated h^+^ in g-C_3_N_4_. This process leaves the high energy h^+^ at the VB of LDHs and the high energy e^−^ at the CB of g-C_3_N_4_ for photocatalysis [44].

To confirm the existence of the type of heterojunction, band structure analysis is crucial [45]. The solid UV-vis absorption and the X-ray photoelectron spectroscopy (XPS) valence band spectrum are common techniques for obtaining the band structure. Through the band structure analysis, we can determine what kind of heterojunction it is (Type I, Type II, and Type III). Notably, Type II and Z-scheme heterojunctions share a similar staggered band alignment structure. Nevertheless, their charge transfer pathways are significantly different. In order to determine if the charge transfer route is Z-scheme or type II heterojunctions, it is therefore critically necessary to investigate it in detail using some techniques such as (i) self-confirmation by photocatalytic reactions, products, and radical species, (ii) selective photodeposition of a noble metal, (iii) in situ XPS, (iv) femto-second transient absorption spectra (fs-TAS), (v) photoassisted Kelvin probe force microscopy (photo-KPFM), theoretical calculations, etc. For example, the work function (Φ) plays a crucial role in investigating the energy band alignment and charge transfer of heterojunctions, making it an essential physical parameter [46]. We can also determine Φ through theoretical calculations and further determine the charge transfer route.

## 3. Synthetic Strategy of LDH/g-C_3_N_4_ Heterojunctions

To obtain good photocatalytic performance, the development of the LDH/g-C_3_N_4_ heterojunctions is crucial. The principal building techniques, such as electrostatic self-assembly, in situ coprecipitation, hydrothermal, solvothermal, and calcination strategies, depend on the assembly strategies of LDHs and g-C_3_N_4_. These synthesis techniques are covered in depth in the sections that follow.

### 3.1. Electrostatic Self-Assembly Method

Layered composites are frequently manufactured via electrostatic self-assembly. It utilizes the electrostatic interactions between materials with various charges [47,48,49]. During the self-assembly process, the electrostatic adsorption with different charges mainly propels the assembly, while electrostatic repulsion between like charges controls the assembly of each layer. In contrast to the electropositive host layer of the LDHs created by the orderly arrangement of metal cations, the suspension of virgin g-C_3_N_4_ in aqueous solution is electronegativity due to –NH_2_ deprotonation [50,51]. By means of electrostatic self-assembly, LDH/g-C_3_N_4_ heterojunctions can be produced thanks to these characteristics. LDHs and g-C_3_N_4_ are exfoliated to nanosheets during the synthesis process before self-assembly. There are numerous ways to exfoliate LDHs using mechanical stirring or ultrasonic treatment [52]. For example, Tahir and co-workers reported 2D/2D CoAl-LDH/g-C_3_N_4_ heterojunctions with a strong interface through the self-assembled deposition of CoAl-LDH flakes onto layered g-C_3_N_4_ nanosheets (Figure 2) [53]. They further used the CoAl-LDH/g-C_3_N_4_ product for MDR and methane bi-reforming to reduce flared gas (methane) with CO_2_. Mg-Al-LDH/g-C_3_N_4_ materials were created by Xu and co-workers using an electrostatic self-assembled method. In this work, the scientists initially produced Mg-Al-LDH through deposition with NaOH and g-C_3_N_4_ from urea by thermal polymerization. Then, g-C_3_N_4_ nanosheets and Mg-AlLDH nanosheets were created using a combination of the hydrothermal method and sonication exfoliation. The produced g-C_3_N_4_ and Mg-Al-LDH showed zeta potentials of 27.3 mV and 52.7 mV, respectively. The Mg-Al-LDH/g-C_3_N_4_ photocatalyst was produced by immediately combining the two photocatalysts, which caused the g-C_3_N_4_ and Mg-Al-LDH to electrostatically interact with each other. The scientists used a transmission electron microscope (TEM) to examine the photocatalyst morphology in order to confirm the 2D/2D assembly of Mg-Al-LDH and g-C_3_N_4_ and discovered that Mg-Al-LDH flakes were evenly distributed onto g-C_3_N_4_ sheets.

### 3.2. In Situ Coprecipitation Strategy

By introducing a precipitating agent, the synchronous precipitation of more than one cation in a uniform solution is known as coprecipitation. Because of the straightforward manipulation, cheap and controllable reaction circumstances, and homogeneous composition, this technology has been a significant means of preparing composites with two or multiple metals [10,54,55]. The process of coprecipitation is also frequently utilized to create LDHs [56]. By combining the metal cations needed for the host layer with alkaline liquor, and then allowing the suspension to age, the necessary LDHs can be produced. The interlayer anions of the LDHs are present in the mixed cationic solution. By adjusting the reaction parameters, e.g., solution pH, temperature, and aging time, the LDH size can be tailored. The basic process for generating LDH/g-C_3_N_4_ through coprecipitation involves depositing the metal cations on the g-C_3_N_4_ nanosheet once they have been electrostatically deposited onto it.

By using in situ coprecipitation, Li and colleagues [57] have successfully created a 2D/2D Zn-Cr-LDH/g-C_3_N_4_ heterojunction. Figure 3 depicts the precise synthesis procedure (from I to V). The modified g-C_3_N_4_ sheets, also known as g-C_3_N_4_-C(N) sheets, were originally created by the authors by combining urea with citric acid. They were then formed into a suspension. Through electrostatic attraction, Zn^2+^ and Cr^3+^ were deposited onto the g-C_3_N_4_-C(N) nanosheet during this procedure. After that, NaOH was used for depositing Zn^2+^ and Cr^3+^ to create Zn-Cr-LDH on the surface of g-C_3_N_4_. The photocatalytic activity of the Zn-Cr-LDH/g-C_3_N_4_-C(N) product used by the authors to further photocatalyze the breakdown of Congo red was obviously higher than Zn-Cr-LDH or g-C_3_N_4_ alone. An in situ crystallization technique, used to create Zn-AlLDH/g-C_3_N_4_ composites, was disclosed by Yuan and co-workers [58]. In this work, Zn^2+^ and Al^3+^ coprecipitated to in situ produce Zn-Al-LDH crystals on g-C_3_N_4_ nanosheets. The microstructure of Zn-Al-LDH/g-C_3_N_4_ was studied by using TEM, and they discovered the comparatively large g-C_3_N_4_ nanosheets were homogeneously coated on Zn-Al-LDH flakes.

### 3.3. Hydrothermal Strategy

The hydrothermal strategy is a popular method to create composites at high temperatures and high pressures in a pressure-tight reactor by using water as a solvent [59,60,61,62]. The fundamental benefit of this technology is the ease with which a crystalline product can be produced via a straightforward hydrothermal procedure. Controlling the reaction conditions also makes it easy to develop the morphology and structure of materials [63]. The process of hydrothermal oxidation, reduction, precipitation, breakdown, polymerization, and so forth are further subcategories of the hydrothermal method depending on the type of reaction that occurs. Numerous researchers have employed the hydrothermal precipitation approach to create LDH/g-C_3_N_4_ heterostructures. Under mild temperature and pressure circumstances, it can be challenging for some metal cations to coprecipitate and produce layered hydroxides, but in a hydrothermal environment with high temperature and pressure, the reaction is simpler to carry out. Additionally, the LDH/g-C_3_N_4_ products that are produced typically have a nice 2D/2D morphology.

By using a hydrothermal process, Liu and colleagues [64] have created a Zn-Cr-LDH/g-C_3_N_4_ composite (Figure 4). In their research, bulk g-C_3_N_4_ was made via thermal polymerization using urea as precursors, and was treated with ultrasound to produce a g-C_3_N_4_ nanosheet suspension. The suspension was then mixed with Zn^2+^, Cr^3+^, and lye. The hydrothermal reaction was carried out for 24 h under 120 °C. The precipitates from the autoclave were eventually collected and dried to produce the Zn-Cr-LDH/g-C_3_N_4_. Chew and co-workers [65] employed urea and NH_4_F instead of NaOH lye for the hydrothermal preparation of Co-Al-LDH/O-doped g-C_3_N_4_ in order to create LDH/g-C_3_N_4_ with uniform 2D/2D morphology. In order to prevent the nonuniform precipitant dispersion and response rate from causing unequal LDH sizes, Co^2+^ and Al^3+^ were precipitated in this technique using the hydrolysis of urea. The hydrolysis of urea produced NH_3_ and CO_2_. While the release of CO_2_ served to agitate the reaction mixture, the produced NH_3_ raised the pH value. As a result, Co-Al-LDH flakes eventually precipitated in a homogenous fashion, further producing high-purity products of consistent size.

### 3.4. Solvothermal Method

The hydrothermal process is improved upon by the solvothermal method. Organic solvents are used as the reaction medium. Although the hydrothermal approach has numerous benefits, it is only capable of producing a small number of non-oxides, such as carbides, nitrides, and phosphides, because both the reactants and the products can hydrolyze or react with H_2_O [32,66,67]. Nonaqueous solvents can be used to properly execute these reactions. Additionally, under high-pressure conditions, several organic solvent properties, such as density, viscosity, and surface tension, fluctuate greatly, which can create specialized media for numerous chemical processes [68]. Due to some organic solvents’ lower boiling points, the gas pressure in solvothermal systems can be higher than that in hydrothermal systems at the same temperature, and the higher pressure encourages the crystallization of the product [69]. When building the LDH/g-C_3_N_4_ heterostructure, an organic solvent can increase the reaction precursors’ dispersity (for instance, a g-C_3_N_4_ suspension), which boosts their chemical reactivity and makes it easier to build 2D/2D structures. Given the benefits, the solvothermal approach was frequently used in investigations to create LDH/g-C_3_N_4_ photocatalysts.

Recently, Zhao and co-workers [70] have created Zn-Al-LDH/g-C_3_N_4_ composites using the solvothermal technique (Figure 5). Ethylene glycol (EG) was used as the reaction environment in their research. Through calcining the urea and suspending it in EG with NaOH, g-C_3_N_4_ was prepared. The solvothermal procedure then involved combining the two EG suspensions. The TEM observation showed that g-C_3_N_4_ sheets were evenly distributed in EG and that the Zn-Al-LDH/g-C_3_N_4_ had a suitable lamellar structure. In addition, it was discovered that the Zn-Al-LDH interlayer spacing (1.03 nm) was greater than ordinary LDHs intercalated with carbonate (0.73 nm). In order to boost photocatalytic efficiency, a comparatively large interlayer spacing can offer larger room for reactant transport and more sites that are active for photosynthesis [71]. Lian and co-workers [72] synthesized a 2D/2D NiCo-LDH/g-C_3_N_4_ Z-scheme heterojunction through the solvothermal technique, which displayed improved photocatalytic efficacy under visible light illumination for the breakdown of tetracycline hydrochloride (TC) and the formation of hydrogen (H_2_). For the NiCo-LDH/g-C_3_N_4_, a Z-scheme mechanism was suggested, demonstrating the heterojunction’s twin benefits of strong redox capacity.

### 3.5. Calcination Method

A substance is heated during calcination to cause it to evaporate H_2_O. This technique is used to create composites of calcined LDH and g-C_3_N_4_. LDHs are utilized as starting materials in the synthesis process, where they are calcined to create mixed metal oxides (MMOs) through geometric modification [73,74]. The produced MMOs exhibit excellent thermal stability and can be widely distributed. The production of metal oxides with an increased specific surface area and porosity allows for the further enhancement of photocatalytic efficiency [75]. Furthermore, calcining LDHs can create spinels, which can improve their capacity to capture visible light [76]. The literature that is currently accessible indicates that there are three calcination techniques to produce a calcined LDH/g-C_3_N_4_ heterojunction. First, the produced LDH/g-C_3_N_4_ composites are immediately calcined. The second technique involves first calcining LDH to MMO and then calcining the compounds of MMO and g-C_3_N_4_ [77]. The third strategy, which was more often employed, involves co-calcining the raw materials for g-C_3_N_4_ and LDH [31]. Figure 6 illustrates the synthesis of dual-S-scheme g-C_3_N_4_/Ti_3_C_2_T/Co_2_Al_0.95_La_0.05_-LDH composites through a simple calcination method [78]. Benefits of the g-C_3_N_4_/Ti_3_C_2_T/CoAlLa-LDH dual-S-scheme assembly include faster charge transfer between the conductor and semiconductor. Additionally, a simple coprecipitation technique was used to create a nanocomposite with a noteworthy 2D/2D heterojunction made of CoFe-LDH loaded on g-C_3_N_4_ nanosheets [79]. When compared to pure g-C_3_N_4_ and CoFe-LDH alone, the prepared nanocomposite showed noticeably higher photocatalytic performance for TC removal.

Numerous preparation techniques, including electrostatic self-assembly, in situ coprecipitation, hydrothermal, solvothermal, and calcination, are beneficial for achieving a close and robust interface between g-C_3_N_4_ and LDH, leading to improved photocarrier migration. Powerful interactions between them, such as covalent bonds, ionic bonds, coordination bonds, hydrogen bonds, and van der Waals forces, may eventually form and serve as pathways for charge transfer while enhancing interface interaction. Especially with the electrostatic self-assembled approach, the original architecture may be conserved and a closely touching interface can be formed. They can be guaranteed to have a homogeneous architecture and small size distribution by using a self-assembly technique. On the other hand, the calcination approach results in a low photocatalytic activity due to the comparatively inadequate interface contact between them. But it is simple and cheap and more suitable for actual industrial production (Table 1).

## 4. LDH/g-C_3_N_4_ Heterojunctions for Organic Pollutant Removal

Because g-C_3_N_4_ and LDHs can create a heterojunction efficiently, it is an unusual approach to increase photocatalytic performance. This is because the as-produced electron-hole pairs are effectively separated. The property of LDH/g-C_3_N_4_ photocatalysts can be significantly enhanced via a rational 2D/2D structure design, allowing for their widespread usage in photocatalytic organic pollution removal. The photocatalytic performance and mechanism of several LDHs/g-C_3_N_4_ are examined and addressed in the section that follows.

Divalent cations such as Co^2+^, Mg^2+^, Ni^2+^, Zn^2+^, etc. as well as trivalent cations such as Al^3+^, Cr^3+^, Fe^3+^, Mn^3+^, Ti^3+^, etc. were employed to build LDH/g-C_3_N_4_ heterojunctions, according to the existing literature. These LDH/g-C_3_N_4_ heterojunctions with improved photocatalytic performance were explored for organic pollutant removal. For instance, the ZnAl-LDH/g-C_3_N_4_ photocatalyst was described by Shanker and colleagues [80] using a microwave-assisted technique. Compared to bare g-C_3_N_4_ and ZnAl-LDH photocatalysts, the optimized ZnAl-LDH/g-C_3_N_4_ compound showed the greatest photodegradation rate constant of 1.22 × 10^−2^ min^−1^ for ciprofloxacin (CIP). Khataee and co-workers [81] reported a g-C_3_N_4_/ZnFe LDH binary heterojunction for the photodegradation of tetracycline (TC) with the aid of Oxone. The effective combination of g-C_3_N_4_/ZnFe LDH/Oxone/UV achieved an evidently improved degradation rate for TC. Thus, it can be also effectively used for the removal of other organic pollutants. Using an easy one-step in situ hydrothermal process, mesoporous g-C_3_N_4_/Zn-Ti LDH-laminated van der Waals heterojunctions were effectively produced [82]. A laminated van der Waals heterostructure was successfully generated between the electronegative g-C_3_N_4_ nanocrystal and the electropositive Zn-Ti LDH, thanks to a strong electrostatic attraction between them. As a result, the resulting heterojunctions demonstrated superior photocatalytic performance for eliminating ceftriaxone sodium completely (>97%). This simple method for creating mesoporous g-C_3_N_4_/Zn-Ti LDH-laminated van der Waals heterojunctions provides new information for creating high-performing photocatalytic materials.

Cerium (Ce) has been widely used as a dopant in LDHs/g-C_3_N_4_ because of its low price and outstanding redox performance [83,84]. Xu and co-workers [85] reported various g-C_3_N_4_/Ce-doped MgAl-LDHs via a solvothermal strategy (Figure 7a). g-C_3_N_4_/MgAl_0.80_Ce_0.20_-LDH has a large SSA (52.71 m^2^ g^−1^) and a good separation efficiency of photocarriers, owing to the coupled effect of Ce-doping and g-C_3_N_4_-LDH heterojunctions (Figure 7b,c). As a result, ~49% Congo red (CR) is adsorbed through g-C_3_N_4_/MgAl_0.80_Ce_0.20_-LDH, and the CR degradation efficiency reaches 90% within 180 min (Figure 7d,e). They also found that both photogenerated h^+^ and superoxide radical **·**O^2−^ could obviously improve photocatalytic CR oxidation performance (Figure 7e). A more effective transfer of photocarriers between g-C_3_N_4_ and MgAl_0.80_Ce_0.20_-LDH is likely the cause of the enhanced photocatalytic property.

Dai and co-workers [41] reported g-C_3_N_4_@Ni-Ti LDH nanocomposites with a high SSA through hydrothermal strategy. These nanocomposites were utilized for the sonophotocatalytic removal of amoxicillin (AMX). The results showed that when AMX is photocatalytically degraded under visible light irradiation, g-C_3_N_4_@Ni-Ti LDH nanocomposites perform better than their individual g-C_3_N_4_ and Ni-Ti LDH. He and his co-workers [86] prepared ZnM-LDH/g-C_3_N_4_ (M = Al, Cr) Z-scheme heterojunctions through the electrostatic self-assembled strategy. ZnAl-LDH and ZnCr-LDHs/g-C_3_N_4_ may be desulfurized by photocatalytic oxidation/extraction in 3 h for model oil, with a 99.8 and 96.6% desulfurization rate, respectively. Mahjoub and co-workers [87] reported a new Bi-doped NiAl-LDH/g-C_3_N_4_ 2D/2D heterojunction for the effective photodegradation of ciprofloxacin (Cipro). Under visible light irradiation, the optimal composite (40%-g-C_3_N_4_/LDH) demonstrated 86% Cipro elimination effectiveness within 180 min. The enhanced flow of photocarriers to the surface and close face-to-face contact between two semiconductors in heterojunction were the key causes of the aforementioned boost in catalytic activity. This method offers a fresh viewpoint on creating 2D/2D heterojunctions between LDH and other 2D semiconductors for the simultaneous detection and photodegradation of antibiotics.

ZnAl-LDH and a non-metal, boron immersion have been used to structurally modify g-C_3_N_4_ in order to increase its light absorption. The obtained 40% B-g-C_3_N_4_ and 30%ZnAl-LDH/g-C_3_N_4_ displayed an increased SSA of 14.3137 and 26.292 m^2^/g, illustrating 90.25 and 86.31% phenol photodegradation within 270 min [88]. Lin and co-workers [79] reported a CoFe-LDH/g-C_3_N_4_ with a notable 2D/2D heterojunction via a simple co-precipitation strategy. When compared to pure g-C_3_N_4_ and CoFe-LDH, the CoFe-LDH/g-C_3_N_4_ nanocomposite showed noticeably higher catalytic performance toward TC photodegradation. Jiao and co-workers [89] reported halogen-doping (F and Cl) g-C_3_N_4_-modified ZnAl-LDH (FCCN/LDH) nanocomposites via a simple coprecipitation process for the photodegradation of TC. In contrast to single FCCN and LDH, the FCCN/LDH showed improved photocatalytic activity, which is because of the proper band alignment and a Z-scheme migration mechanism. The tenable Z-scheme mechanism indicated that the primary free radicals involved in photodegradation are h^+^ and **·**O^2−^ active species (Figure 8). In addition, flower-like g-C_3_N_4_/NiZnAl-LDH S-scheme heterojunctions with oxygen vacancies [90] and vulcanized ZnAl LDH-modified g-C_3_N_4_ heterojunctions [91] also displayed improved photocatalytic performance.

The photocarriers’ separation can be increased by connecting LDH and g-C_3_N_4_, yet some LDH/g-C_3_N_4_ photocatalysts have charge carriers that are difficult to further migrate and engage in redox processes [92]. Building LDH/g-C_3_N_4_/X ternary photocatalysts, therefore, needs to be taken into consideration, where X stands for another photocatalyst or noble metal. This plan is anticipated to improve light harvesting and the transfer of photocarriers. Ag and reduced graphene oxide (RGO) were predominantly utilized to achieve these targets, according to the research that is currently available. For instance, Fazaeli et al. [93] prepared a new Ag-bridged dual Z-scheme Ag/g-C_3_N_4_/CoNi-LDH plasmonic heterojunction by a simple hydrothermal method. The most favorable degradation conditions for a TC solution were found to be 7 pH, 40 g L^−1^ catalyst dosage, 30 mg L^−1^ TC concentration, and a 100 min irradiation period. Pazhanivel and co-workers [94] reported an RGO-supported g-C_3_N_4_/NiMgAl LDH hybrid for achieving high-performance photocatalytic organic pollution removal. A ternary heterojunction consisting of CoAl-LDH, g-C_3_N_4_, and RGO (LDH/CN/RGO) with a noticeable 2D/2D/2D structure utilizing a facile hydrothermal strategy was reported recently [92] (Figure 9a). Of all the manufactured catalysts, the LDH/CN/RGO ternary heterojunction with RGO and LDH contents of 1 weight percent and 15 weight percent, respectively, displayed the highest degrading efficiency (Figure 9b,c). It also showed outstanding stability throughout recycling studies. Due to the unique 2D/2D/2D configuration of the constituent CN, LDH, and RGO, speeding up the photocarriers’ transport processes to effectively impede their recombination resulted in enhanced photocatalytic performance (Figure 9d,e). Furthermore, the possible photodegradation routes for both CR and TC were proposed on the basis of the intermediate products (Figure 9f). Therefore, with their high performance and ability to respond to visible light, the current LDH/CN/RGO ternary heterojunctions show great potential for real-world uses in solar energy transformation and environmental preservation.

Besides the above common LDH/g-C_3_N_4_ heterojunctions, researchers have also made great efforts to develop other candidates to prepare LDH/g-C_3_N_4_ heterojunctions, e.g., NiFe-LDH/g-C_3_N_4_ [95], g-C_3_N_4_@NiAl-LDH [96], g-C_3_N_4_@NiFe-LDH [97], g-C_3_N_4_/MgZnAl-calcined LDH [98], ZnMgAl LTH/ZnO/g-C_3_N_4_ [99], Co-Al LDH/g-C_3_N_4_-CoFe_2_O_4_ [100], g-C_3_N_4_/ZnFeMMO [76], g-C_3_N_4_@LDH/NCQDs [101], NiFe-LDH/NRGO/g-C_3_N_4_ [102], ZnCr-LDH/g-C_3_N_4_-C(N) [57], In_2_S_3_/g-C_3_N_4_/CoZnAl-LDH [103], ZnTi/C_3_N_4_/Ag LDH [104], etc. As exhibited in Table 2, g-C_3_N_4_@Ni-Ti LDH [41] displayed a very high photocatalytic activity by sonophotocatalytic process, and the proposed sonophotocatalytic process and the relatively low cost give them huge application prospects. Furthermore, multicomponent LDH/g-C_3_N_4_-based photocatalysts such as CoAl-LDH/g-C_3_N_4_/RGO [92] displayed some unique merits, and thus it is more worthy of attention. Moreover, for more clarification and improvement, a deeper understanding of the charge migration pathway and the photocatalytic mechanism is essential. Beyond question, LDH/g-C_3_N_4_ systems will have huge potential in photocatalytic organic pollutant removal.

## 5. Conclusions and Outlook

In conclusion, developing LDH/g-C_3_N_4_ heterojunctions is a successful method for achieving effective photocatalysis for the removal of organic pollution. The separation and transfer of photoinduced charge carriers can be substantially facilitated by the planar configuration and close contact of LDH and g-C_3_N_4_, which improves photocatalytic performance. For building an LDH/g-C_3_N_4_ heterojunction with the 2D/2D and suitable band structures, a variety of synthesis techniques have been reported, including electrostatic self-assembly, in situ coprecipitation, hydrothermal, solvothermal, and calcination strategies. An excellent photocatalytic property in the breakdown of organic pollutants may be attributed to the synergistic interaction between LDH and g-C_3_N_4_. The research investigations on LDH/g-C_3_N_4_ photocatalysts are still in their early stages, despite the many successes that have been made, and it is clear that more study is required. The following examples of opportunities and difficulties call for greater emphasis and investigation:(i)Matching the redox potential of a particular photocatalytic process with the CB (or VB) potential of LDH/g-C_3_N_4_ photocatalysts is crucial. This is likely to modify the CB (or VB) position of LDHs/g-C_3_N_4_ to offer high redox potential for a variety of photocatalytic processes since the band structure of the compound is customizable. The precise adjustment technique, however, requires more research.(ii)Gaining an in-depth understanding of the transfer mechanism of charge is also crucial. The precise charge transfer pathway and photocatalytic process can be further provided using theoretical calculations and characterization methods [61,105,106] including in situ FTIR, in situ XPS, photo-KPFM, and synchrotron radiation. The search for additional LDH can combine with g-C_3_N_4_ to improve photocatalytic activity, which is made easier with a greater knowledge of the mechanism.(iii)Constructing ultrathin 2D/2D structures is highly worth considering. The ultrathin LDHs and g-C_3_N_4_ have been suggested as photocatalysts in certain investigations. This is also likely to construct ultrathin LDH/g-C_3_N_4_ heterojunctions that will enable more rapid photocarrier migration because of the further shortened migration distance and decreased photocarrier recombination.(iv)Enhancing the capacity to capture visible or even near-infrared light, which accounts for over half of solar radiation, is strongly advised. Even while LDH/g-C_3_N_4_ photocatalysts can work with visible light, surface sensitization, doping, band gap correction, and other techniques can help them gather sunlight more effectively.(v)Expanding the applications of LDH/g-C_3_N_4_ photocatalysts in light of their special advantages is essential to creating a sustainable society.(vi)Some effective enhancement strategies, such as morphological modulation, loading co-catalysts, interface engineering, doping, and exposing more reactive facets, should be seriously studied to achieve an actual application of LDH/g-C_3_N_4_ systems.(vii)Improving the ability to harness both visible light and near-infrared light, which collectively represent more than 50% of solar irradiation, is paramount. LDH/g-C_3_N_4_ photocatalysts show potential for efficient operation under visible light, and there are also opportunities to further enhance sunlight harvesting capacity through techniques such as surface sensitization, doping elements, adjusting the band gap, and more.

Looking ahead, this developing field offers both many prospects and difficult problems. 

This review is thought to be able to offer some fresh and sophisticated perspectives for directing the logical design of LDH/g-C_3_N_4_ systems for organic pollution removal. There is no doubt that the LDH/g-C_3_N_4_ systems will soon have useful industrial applications.

## Figures and Tables

**Figure 1 nanomaterials-13-03066-f001:**
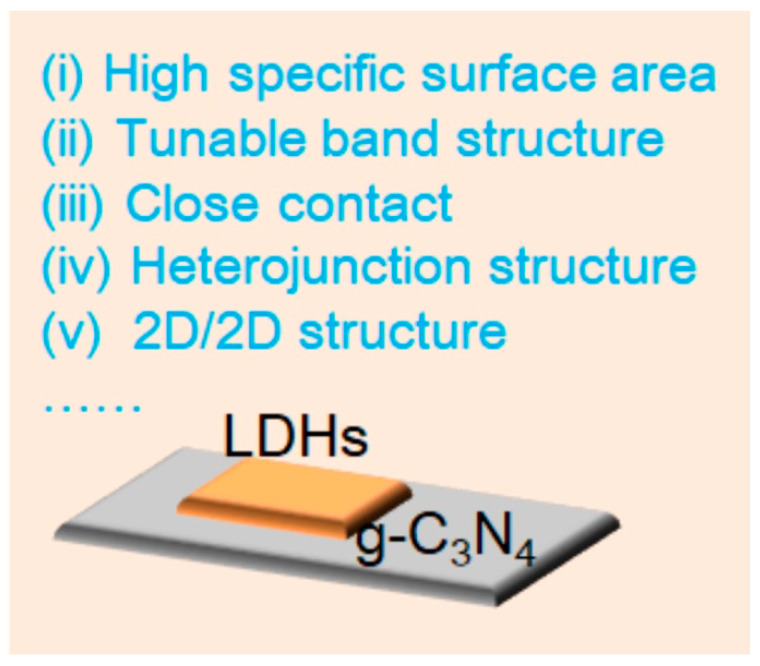
Advantages of LDH/g-C_3_N_4_ heterojunctions.

**Figure 2 nanomaterials-13-03066-f002:**
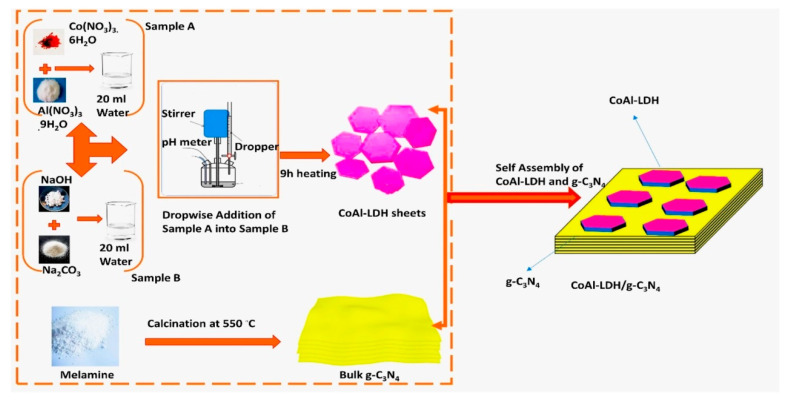
Schematic illustration of the synthesis of CoAl-LDH, g-C_3_N_4_, and CoAl-LDH/g-C_3_N_4_ composite samples. Reproduced with permission [53].

**Figure 3 nanomaterials-13-03066-f003:**
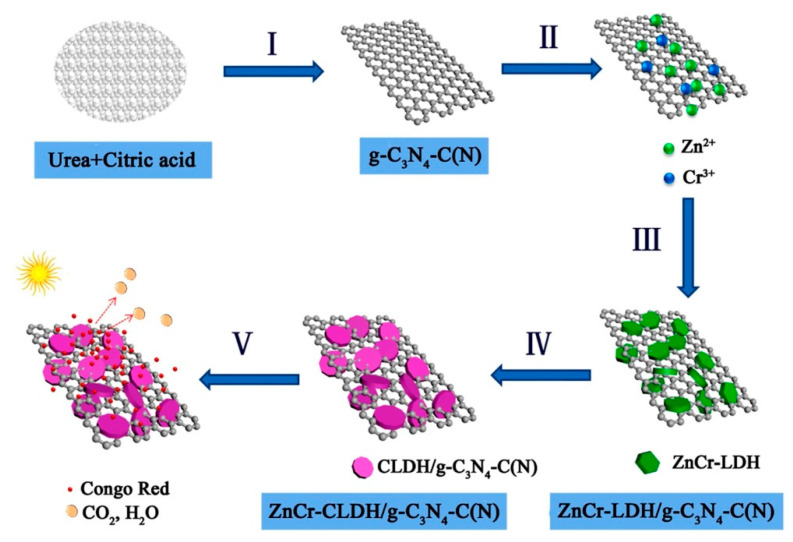
Diagrammatic sketch of the synthesis process and photocatalytic property of ZnCr-CLDH/g-C_3_N_4_-C(N). (I) Thermal polymerization at 550 °C, (II) introduction of Zn^2+^ and Cr^3+^ under the condition of continuous stirring, (III) in situ precipitation of ZnCr-LDH on g-C_3_N_4_-C(N), (IV) calcination and then formation of ZnCr-CLDH/g-C_3_N_4_-C(N); (V) adsorption and photocatalytic performance for CR removal under visible light irradiation. Reproduced with permission [57].

**Figure 4 nanomaterials-13-03066-f004:**
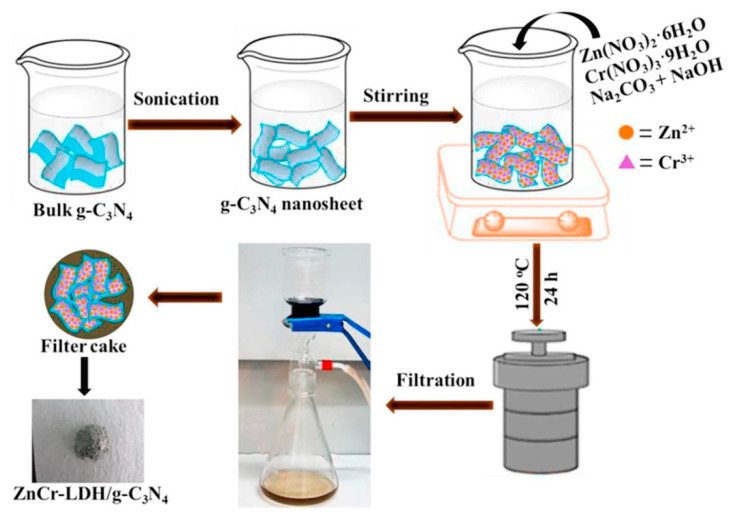
Representative synthesis process of ZnCr-LDH/g-C_3_N_4_. Reproduced with permission [64].

**Figure 5 nanomaterials-13-03066-f005:**
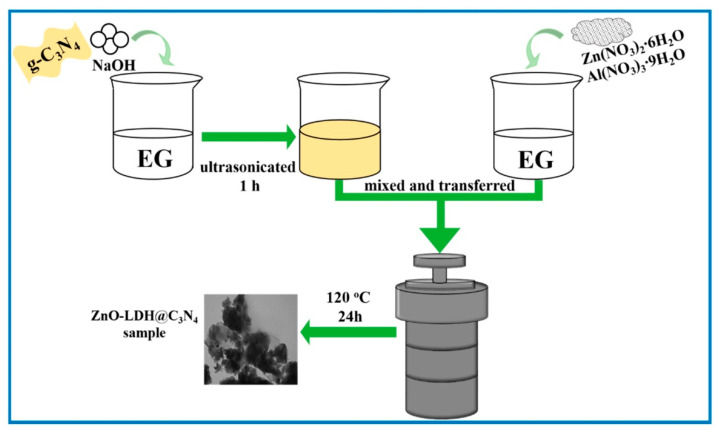
The representative synthesis process of Zn-Al-LDH/g-C_3_N_4_. Reproduced with permission [70].

**Figure 6 nanomaterials-13-03066-f006:**
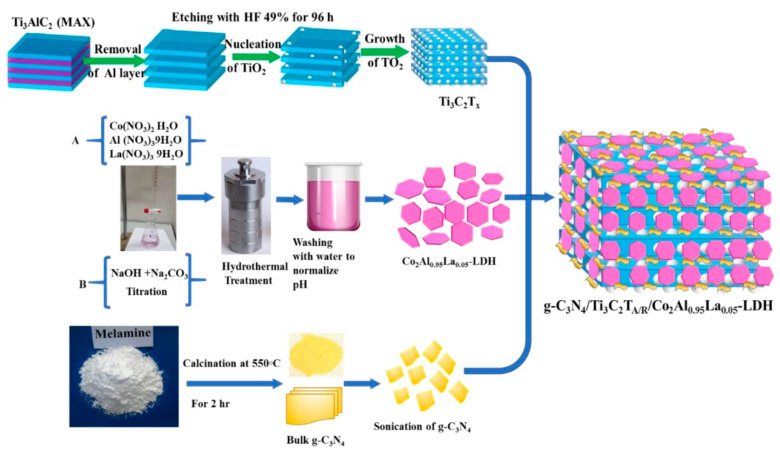
Schematic representation of the synthesis process of g-C_3_N_4_/Ti_3_C_2_T/Co_2_Al_0.95_La_0.05_-LDH composite. Reproduced with permission [78].

**Figure 7 nanomaterials-13-03066-f007:**
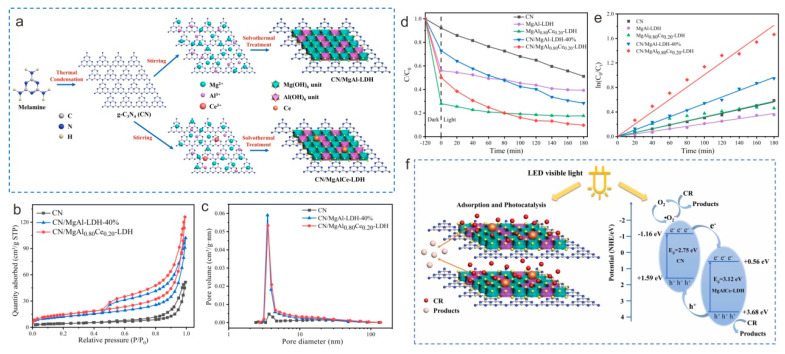
(**a**) Diagrammatic sketch of fabrication route of g-C_3_N_4_/MgAl_0.80_Ce_0.20_-LDH. BJH N_2_ adsorption-desorption isotherms (**b**) and pore size distribution (desorption) (**c**) of g-C_3_N_4_ and g-C_3_N_4_/Ce-doped MgAl-LDHs. CR adsorption capacity (**d**) and pseudo-first-order kinetic plots for CR photodegradation (**e**) of g-C_3_N_4_ and g-C_3_N_4_/Ce-doped MgAl-LDHs. (**f**) Mechanism explanation of photocatalytic degradation via g-C_3_N_4_/Ce-doped MgAl-LDHs. Reproduced with permission [85].

**Figure 8 nanomaterials-13-03066-f008:**
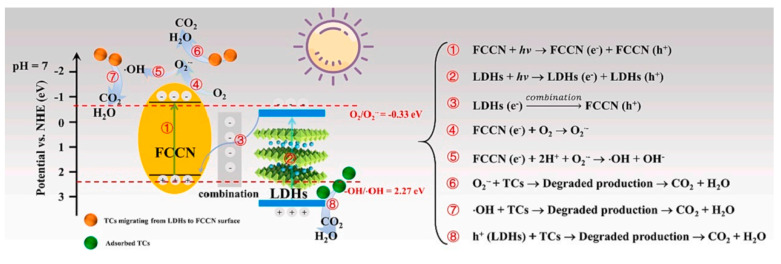
Diagrammatic sketch of FCCN/LDH-100 nanocomposites under visible light illumination. Reproduced with permission [89].

**Figure 9 nanomaterials-13-03066-f009:**
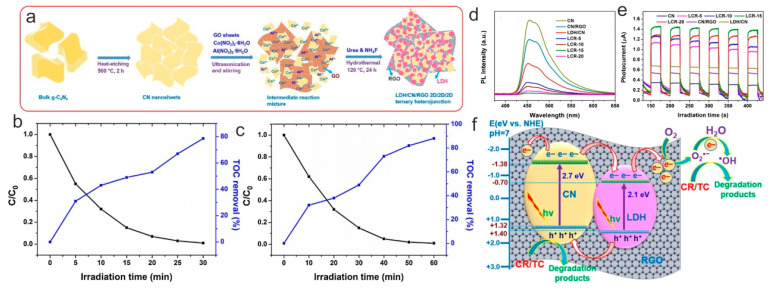
(**a**) Diagrammatic sketch of the preparation of LDH/CN/RGO 2D/2D/2D heterojunctions. Comparison of the photocatalytic activity and TOC removal rate over the LCR-15 photocatalyst in the photodegradation of CR (**b**) and TC (**c**). PL spectra (**d**) and photocurrent responses (**e**) of CN, LDH/CN, CN/RGO, and LDH/CN/RGO photocatalysts. (**f**) Mechanism explanation of photocatalytic degradation of CR and TC over LDH/CN/RGO. Reproduced with permission [92].

**Table 1 nanomaterials-13-03066-t001:** Comparison between different methods for the preparation of LDH/g-C_3_N_4_ heterojunctions.

Synthesis Strategy	Advantages	Disadvantages
Electrostatic self-assembly method	Very suitable for the preparation of 2D/2D LDHs/g-C_3_N_4_	Time- and cost-consuming, complicated preparation process, and low efficiency
In situ coprecipitation method	Facile, eco-friendly, and highly efficient	Easy to agglomerate
Hydrothermal and solvothermal method	(i) Facile, eco-friendly, and highly efficient (ii) Produced photocatalysts with a relatively high crystallinity, small size distribution, and controllable architecture	Complex preparation process
Calcination method	Facile, eco-friendly, and effective, and reduced aggregation	Relatively poor interface contacts between two semiconductors

**Table 2 nanomaterials-13-03066-t002:** Comparison of the photocatalytic organic pollution removal property over LDH/g-C_3_N_4_ systems.

Photocatalysts	Preparation Strategy	Mass (mg)	Light Source	Target Pollutant/Initial Concentration	Degradation Time (min) and Rate (%)	*K_app_ ^a^* [10^−2^ min^−1^]	Ref.
ZnAl-LDH/ g-C_3_N_4_	Microwave-assisted method	30	35 W Xe arc lamp	CIP/20 mg L^−1^	140 (~84.1)	1.22	[80]
g-C_3_N_4_/ZnFe LDH	Mechanical stirrer method	50–300	6-W UVC lamp	TC/10–55 μM	30 (~92.4)	-	[81]
g-C_3_N_4_/Zn-Ti LDH	Hydrothermal method	100	300 W xenon lamp, >420 nm	Ceftriaxone sodium/10 mg L^−1^	240 (~97)	1.14	[82]
g-C_3_N_4_/ MgAl_0.80_Ce_0.20_-LDH	Solvothermal method	20	5 W LED lamp, 400–760 nm	CR/50 mg L^−1^	180 (~90)	1.01	[85]
g-C_3_N_4_@Ni-Ti LDH	Hydrothermal method	-	400 W Hg lamp	AMX/1000 mg L^−1^	75 (~99.5)	-	[41]
ZnM-LDH/ g-C_3_N_4_ (M = Al, Cr)	Electrostatic self-assembly method	180	500 W mercury lamp	Model oil/90 mL	180 (~99.8)	-	[86]
Bi-doped NiAl-LDH/ g-C_3_N_4_	Annealing method	50	400 W Mercury-vapor lamp, >400 nm	Cipro/10 mg L^−1^	180 (~86)	-	[87]
ZnAl-LDH/ g-C_3_N_4_	Thermal condensation	-	-	Phenol/700 mg L^−1^	270 (~62.38)	-	[88]
CoFe-LDH/ g-C_3_N_4_	Coprecipitation method	-	5 W LED light, >420 nm	TC/40 mg L^−1^	180 (~83.8)	-	[79]
FCCN/LDH	Coprecipitation process	50	300 W Xe lamp	TC/20 mg L^−1^	120 (~72.13)	2.314	[89]
g-C_3_N_4_/ NiZnAl-LDH	Hydrothermal method	25	500 W Xenon lamp, >400 nm	TC/10 mg L^−1^	120 (>99)	2.329	[90]
ZnAlS*_x_*@ g-C_3_N_4_	Hydrothermal method	50	300 W Xe lamp	TC/20 mg L^−1^	180 (~94.05)	-	[91]
Ag/g-C_3_N_4_/ CoNi-LDH	Hydrothermal method	40	100 W Xe lamp	TC/30 mg L^−1^	100 (~86.3)	-	[93]
RGO/g-C_3_N_4_/ NiMgAl LDH	Hydrothermal method	20	250 W mercury lamp	MB/50 mg L^−1^	75 (~95.14)	1.8	[94]
LDH/CN/RGO	Hydrothermal method	50	300-W halogen lamp	TC/20 mg L^−1^	30 (>99)	-	[92]
NiFe-LDH/ g-C_3_N_4_	Hydrothermal method	20	500 W Xenon lamp, >420 nm	-	60 (~96.81)	5.457	[95]
g-C_3_N_4_@ NiAl-LDH	Hydrothermal method	600	500W high-pressure Hg lamp.	-	180 (~99)	-	[96]
g-C_3_N_4_@ NiFe-LDH	Hydrothermal method	50	500 W Xenon lamp, >420 nm	-	240 (~99)	1.52	[97]
g-C_3_N_4_/MgZnAl-calcined LDH	Template method	250	300 W Xenon lamp, >420 nm	-	240 (>99)	-	[98]
ZnMgAl LTH/ ZnO/g-C_3_N_4_	Stirring strategy	10	300 W Xenon lamp	MB/50 mg L^−1^	75 (>99)	-	[99]
Co-Al LDH/g-C_3_N_4_- CoFe_2_O_4_	Stirring strategy	-	50 W LED lamp	-	-	2.4	[100]
g-C_3_N_4_/ ZnFeMMO	Stirring strategy	25	500 W Xenon lamp, >300 nm	Sulfadiazine/5 mg L^−1^	240 (~96.4)	1.317	[76]
g-C_3_N_4_@LDH/NCQDs	Hydrothermal method	100	300 W Xenon lamp, >400 nm	TC/20 mg L^−1^	120 (~90)	-	[101]
NiFe-LDH/ NRGO/g-C_3_N_4_	Calcination- electrostatic self-assembly method	20	-	RhB/20 mg L^−1^	120 (~97)	-	[102]
ZnCr-LDH/ g-C_3_N_4_-C(N)	Coprecipitation method	50	500 W Xenon lamp, >400 nm	CR/20 mg L^−1^	60 (~70)	1.924	[57]
In_2_S_3_/g-C_3_N_4_/ CoZnAl-LDH	Hydrothermal method	150	300 W Xe lamp, 320–780 nm	MO/50 mg L^−1^	120 (~90.75)	-	[103]
ZnTi/C_3_N_4_/ Ag LDH	Self-assembly method	150	300 W Xe lamp	Phenol/20 mg L^−1^	210 (~76.6)	-	[104]

*^a^ K_app_* represents the reaction rate constant.

## Data Availability

The data presented in this study are available in this article.

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
