# Peer review of "Recent Advances in LDH/g-C3N4 Heterojunction Photocatalysts for Organic Pollutant Removal"

_nanomaterials, 2023, doi:10.3390/nano13233066_

Round 1

Reviewer 1 Report

Comments and Suggestions for Authors

The manuscript nanomaterials-2698926, entitledRecent advances on LDHs/g-C3N4 heterojunction photocatalysts for organic pollutant removal by Cheng Du et al., deals with a review of several reported works about the preparation and photocatalytic activity of LDHs/g-C3N4 heterojunction photocatalysts. This review is a bit descriptive and some important information is missing. Furthermore, some problems should be addressed before this paper can be considered to be accepted.

1 - The English writing should be improved. There are some repetitions of expressions that can be avoid and typographical and grammatical errors in the manuscript. Hence, the manuscript should be carefully checked, and necessary corrections should be done.

2 – In Figure 3, the arrows from I to V is not perceptible for the reader, this should be identified in the legend. Probably in the original paper the authors mentioned in the text, but in this review should be also mentioned.

3 – This review is about heterojunction, however no mention how the type and prove of the heterojunction formation was referred. This information should be introduced for all the referred papers. Because has been publish some papers saying that they have a heterojunction, however no evidence and proves were provide. So, this is information is missing.

4 – Just one figure (Fig. 8) has the activation mechanism of the photocatalysts, should be interesting to have this for all the papers used for this review paper.

Comments on the Quality of English Language

The English writing should be improved, there are some repetitions of expressions that can be avoid.

Author Response

The manuscript nanomaterials-2698926, entitled “Recent advances on LDHs/g-C3N4 heterojunction photocatalysts for organic pollutant removal” by Cheng Du et al., deals with a review of several reported works about the preparation and photocatalytic activity of LDHs/g-C3N4 heterojunction photocatalysts. This review is a bit descriptive and some important information is missing. Furthermore, some problems should be addressed before this paper can be considered to be accepted.

Reply: Thank for your constructive comments. We have revised it well according to your suggestions. We are confident that the new version can meet the high confident of Nanomaterials. The following section is a point-to-point response.

(1) - The English writing should be improved. There are some repetitions of expressions that can be avoid and typographical and grammatical errors in the manuscript. Hence, the manuscript should be carefully checked, and necessary corrections should be done.

Reply: Thank for your constructive comments. English writing has been improved by some professional English teacher. In addition, repetitions of expressions and typographical and grammatical errors have been also revised.

(2) – In Figure 3, the arrows from I to V is not perceptible for the reader, this should be identified in the legend. Probably in the original paper the authors mentioned in the text, but in this review should be also mentioned.

Reply: Thank for your constructive comments. The arrows from I to V has been identified in the legend. And it also has been mentioned in this review.

(3) – This review is about heterojunction, however no mention how the type and prove of the heterojunction formation was referred. This information should be introduced for all the referred papers. Because has been publish some papers saying that they have a heterojunction, however no evidence and proves were provide. So, this is information is missing.

Reply: Thank for your constructive comments. The the type and prove of the heterojunction formation have been provided.

(4) – Just one figure (Fig. 8) has the activation mechanism of the photocatalysts, should be interesting to have this for all the papers used for this review paper.

Reply: Other figures (Fig. 7 and 9) have also added the activation mechanism of the photocatalysts.

Reviewer 2 Report

Comments and Suggestions for Authors

C. Du et al. describe the recent progress in LDHs/g-C3N4 heterojunctions utilized for pollutant removal from contaminated environment. The review is timely and many references are contemporary.

There are some annoyances regarding the use of English language. Various rephrasing is needed, and some words can be replaced by more appropriate terms (avoid for instance “fantastic method” – line 37 ; “reaction was presented” – line 189 etc.)

The quality of the pictures is somewhat low (text can hardly be read on some instances). This needs to be addressed.

The concept and execution of Fig. 1 leave a lot to be desired…

Comments on the Quality of English Language

Careful proofreading is recommended. There are some annoyances regarding the use of English language. Various rephrasing is needed, and some words can be replaced by more appropriate terms (avoid for instance “fantastic method” – line 37 ; “reaction was presented” – line 189 etc.)

Author Response

Du et al. describe the recent progress in LDHs/g-C3N4 heterojunctions utilized for pollutant removal from contaminated environment. The review is timely and many references are contemporary.

Reply: Thank for your constructive comments. We have revised it well according to your suggestions. We are confident that the new version can meet the high confident of Nanomaterials. The following section is a point-to-point response.

(1) There are some annoyances regarding the use of English language. Various rephrasing is needed, and some words can be replaced by more appropriate terms (avoid for instance “fantastic method” – line 37 ; “reaction was presented” – line 189 etc.)

Reply: Thank for your constructive comments. These inappropriate statements have been revised.

(2) The quality of the pictures is somewhat low (text can hardly be read on some instances). This needs to be addressed.

Reply: The quality of the pictures have been improved.

(3) The concept and execution of Fig. 1 leave a lot to be desired…

Reply: The Fig. 1 has been revised.

Reviewer 3 Report

Comments and Suggestions for Authors

The following issues must be addressed:

1.       Introduction part must be significantly improved in order to outline why this review is necessary for the scientific community, as there are already many reviews in this field.

2.       Introduction part should include other alternatives such as heterostructures (i.e. DOI: 10.1016/j.molliq.2023.123384; 10.1016/j.cattod.2017.03.018; 10.1016/j.cej.2023.147347).

3.       Chapter 2 should include both advantages and disadvantages in order to be objective.

4.       Chapter 3 must include a critical approach when comparing the data; it is essential to include personal opinions based on the manuscript authors experience in this field.

5.       All abbreviations must be explained (including abbreviations present in figures).

6.       Some active species must be corrected (authors mistakes the subscript with superscript and vice versa).

7.        It will be useful to maintain the same units measure for pollutant concentration (Table 2) in order to be more easily compared.

8.       Perspectives part should be extended as well.

Author Response

(1)       Introduction part must be significantly improved in order to outline why this review is necessary for the scientific community, as there are already many reviews in this field.

Reply: Thank for your constructive comments. Introduction part has been significantly improved.

(2)       Introduction part should include other alternatives such as heterostructures (i.e. DOI: 10.1016/j.molliq.2023.123384; 10.1016/j.cattod.2017.03.018; 10.1016/j.cej.2023.147347).

Reply: These nice references have cited in Introduction part.

(3)       Chapter 2 should include both advantages and disadvantages in order to be objective.

Reply: The advantages and disadvantages have been added in Chapter 2.

(4)       Chapter 3 must include a critical approach when comparing the data; it is essential to include personal opinions based on the manuscript authors experience in this field.

Reply: My personal opinions have been added in Chapter 3.

(5)       All abbreviations must be explained (including abbreviations present in figures).

Reply: All abbreviations have been explained.

(6)       Some active species must be corrected (authors mistakes the subscript with superscript and vice versa).

Reply: Active species have been corrected.

(7)        It will be useful to maintain the same units measure for pollutant concentration (Table 2) in order to be more easily compared.

Reply: Table 2 has been revised well according to your suggestion.

(8)       Perspectives part should be extended as well.

Reply: Perspectives part has been extended.

Round 2

Reviewer 1 Report

Comments and Suggestions for Authors

The manuscript nanomaterials-2698926 has been improved, however there is still some missing information:

1 – In Figure 3, the arrows (I-V) information is still missing. In the referred paper in reference [55] this is the information in the legend: “Scheme 1. Schematic illustration of the fabrication route and photocatalytic performance of hybrid ZnCr-CLDH/g-C3N4-C(N) nanocomposites. (I) Thermal polymerization at 550 °C; (II) adding Zn2+ and Cr3+ under stirring; (III) in situ precipitation of ZnCrLDH on g-C3N4-C(N); (IV) calcinations and formation of ZnCrCLDH/g-C3N4-C(N); (V) adsorption and photocatalytic CR under visible light.”

The part “(I) Thermal polymerization at 550 °C; (II) adding Zn2+ and Cr3+ under stirring; (III) in situ precipitation of ZnCrLDH on g-C3N4-C(N); (IV) calcinations and formation of ZnCrCLDH/g-C3N4-C(N); (V) adsorption and photocatalytic CR under visible light.” should be included in this review paper.

2 – An explanation how to prove the presence of a heterojunction and not just a mixture of two compounds should be include. What measurements and techniques are necessary to prove the presence of a heterojunction for example should be included.

Author Response

The manuscript nanomaterials-2698926 has been improved, however there is still some missing information:

Reply: Thank for your constructive comments. We have revised it well according to your suggestions. We are confident that the new version can meet the high confident of Nanomaterials. The following section is a point-to-point response.

1 – In Figure 3, the arrows (I-V) information is still missing. In the referred paper in reference [55] this is the information in the legend: “Scheme 1. Schematic illustration of the fabrication route and photocatalytic performance of hybrid ZnCr-CLDH/g-C3N4-C(N) nanocomposites. (I) Thermal polymerization at 550 °C; (II) adding Zn2+ and Cr3+ under stirring; (III) in situ precipitation of ZnCrLDH on g-C3N4-C(N); (IV) calcinations and formation of ZnCrCLDH/g-C3N4-C(N); (V) adsorption and photocatalytic CR under visible light.” The part “(I) Thermal polymerization at 550 °C; (II) adding Zn2+ and Cr3+ under stirring; (III) in situ precipitation of ZnCrLDH on g-C3N4-C(N); (IV) calcinations and formation of ZnCrCLDH/g-C3N4-C(N); (V) adsorption and photocatalytic CR under visible light.” should be included in this review paper.

Reply: Thank for your constructive comments. According to your suggestion, we have added the arrows (I-V) information in this part.

2 – An explanation how to prove the presence of a heterojunction and not just a mixture of two compounds should be include. What measurements and techniques are necessary to prove the presence of a heterojunction for example should be included.

Reply: Thank for your constructive comments. We have added some typical measurements and techniques for proving the presence of a heterojunction.

Reviewer 3 Report

Comments and Suggestions for Authors

The manuscript can be published in present form.

Author Response

The manuscript can be published in present form.

Reply: Thank for your positive comments.

Round 3

Reviewer 1 Report

Comments and Suggestions for Authors

The authors answered all questions and added the requested information. Therefore this manuscript can be accepted as it is for publication in Nanomaterials.

Author Response

Thanks for your positive comments.